# Excellent Photonic and Mechanical Properties of Macromorphic Fibers Formed by Eu^3+^-Complex-Anchored, Unzipped, Multiwalled Carbon Nanotubes

**DOI:** 10.3390/ma15144933

**Published:** 2022-07-15

**Authors:** Mengjie Huang, Haihang Wang, Gaohan Liu, Heng Wei, Jie Hu, Yao Wang, Xuezhong Gong, Sui Mao, Michail Danilov, Ihor Rusetskyi, Jianguo Tang

**Affiliations:** 1Institute of Hybrid Materials, National Centre of International Joint Research for Hybrid Materials Technology, National Base of International Science & Technology Cooperation on Hybrid Materials, Qingdao University, 308 Ningxia Road, Qingdao 266071, China; mojoyellow@163.com (M.H.); whh_yy79@163.com (H.W.); xiaonuanhan@163.com (G.L.); 2021010020@qdu.edu.cn (H.W.); nbbjiehu@126.com (J.H.); wangyaoqdu@126.com (Y.W.); xzgong@163.com (X.G.); maosui001@qdu.edu.cn (S.M.); 2V.I. Vernadskii Institute of General and Inorganic Chemistry of the Ukrainian NAS, 32/34 Palladin Avenue, 03142 Kyiv, Ukraine; r2_igor71@ukr.net

**Keywords:** multiwalled carbon nanotube (MWCNT), Eu^3+^-complex, luminescence, wet-spun fiber

## Abstract

The macromorphic properties of carbon nanotubes perform poorly because of their size limitations: nanosize in diameters and microsize in length. In this work, to realize these dual purposes, we first used an electrochemical method to tear the surface of multiwalled carbon nanotubes (MWCNTs) to anchor photonic Eu^3+^-complexes there. Through the polar reactive groups endowed by the tearing, the Eu^3+^-complexes coordinate at the defected structures, obtaining the Eu^3+^-complex-anchored, unzipped, multiwalled carbon nanotubes (E-uMWCNTs). The controllable surface-breaking retains the MWCNTs’ original, excellent mechanical properties. Then, to obtain the macromorphic structure with infinitely long fibers, a wet-spinning process was applied via the binding of a small quantity of polyvinyl alcohol (PVA). Thus, the wet-spun fibers with high contents of E-uMWCNTs (E-uMWCNT-Fs) were produced, in which the E-uMWCNTs took 33.3 wt%, a high ratio in E-uMWCNT-Fs. On the other hand, due to the reinforcing effect of E-uMWCNTs, the highest tensile strength can reach 228.2 MPa for E-uMWCNT-Fs. Meanwhile, the E-uMWCNT-Fs show high-efficiency photoluminescence and excellent media resistance performance due to the embedding effect of PVA on the E-uMWCNTs. Therefore, E-uMWCNT-Fs can exhibit excellent luminescence properties in aqueous solutions at pH 4~12 and in some high-concentration metal-ion solutions. Those distinguished performances promise outstanding innovations of this work.

## 1. Introduction

Multiwalled carbon nanotubes (MWCNTs) have excellent properties (such as conductivity [1,2], photon absorption in the wide spectrum [3], super-strong mechanical properties [4,5], and special quantum characteristics [6,7]) while being nanosized in diameter and microsized in length [8]. However, the macromorphic properties perform poorly because of their nanosized diameters and microsized lengths [9]. They also exhibit inert reactivity to combined species and poor dispersibility in host media, including solids and liquids, on account of their perfect-layer graphene structure [10]. To explore the macromorphic properties of MWCNTs, the main method is to oxidize their surfaces [11] and then macromorphize them with matrix materials. Strong acids (such as nitric acid [12], sulfuric acid [13], etc.), strong oxidants (such as potassium permanganate [14]) or their mixture [15,16] oxidize the surfaces of the MWCNTs, causing defects on their surfaces and producing oxygen-containing groups. The products derived from the technologies mentioned above are called “unzipped, multiwalled carbon nanotubes” (uMWCNTs). Nevertheless, the abovementioned technologies can only handle a small amount (i.e., 10^−3^ g) of MWCNTs in each batch because of their huge specific volume per gram; meanwhile, those technologies severely destroy the structures of carbon nanotubes [14].

Therefore, the proper technology for producing controllable, slight surface-breaking and retaining their original, excellent mechanical properties is urgently needed. In addition, a reduction in the use of strong acids and bases are required for environmental protection. As an example, Shinde et al. [17] reported an electrochemical method for treating MWCNTs in a sulfuric acid solution to process 10 mg in each batch although the products was characterized by unzipping that was too slight to see the breaking surface of the treated MWCNTs. Through the slight unzipping of MWCNTs, the generated polar groups at the surface of uMWCNTs can endow their reactivity with coming functional components [18]. Then, the huge opportunity to develop a variety of functional macromorphic materials will emerge. They will retain all of the excellent properties of the original carbon nanotubes, including their electronic and mechanical properties [19], and they will be applied to macrosized materials [9,20]. The macromorphic fibers formed by uMWCNTs are most urgently awaited because they can be processed and shaped into practical, high-performance materials in cutting-edge applications [10,21].

Lanthanides are known as "a treasure trove of new materials" due to the optical properties generated by their unique electron orbital structures [22], and the complexes prepared by lanthanide ions exhibit excellent luminescence properties [23]. When the ligand reaches the excitation state under ultraviolet light, the energy absorbed by the ligand is transferred to the lanthanide ions in a non-radiative way, which causes the rare-earth ions to emit strong luminescence [24,25].

In this work, we undertook serious modifications of the electrochemical reactor, including increasing the efficiency of the bath ratio, increasing the packing density of the MWCNTs between the cathode and anode, and obtaining control over uMWCNTs by adjusting the strength of the current and the electrode area. This modified technique not only increased production efficiency, but it also decreased the pollution by strong acids and bases. Furthermore, uMWCNTs can provide the proper reactivity needed to anchor Eu^3+^-complexes at their defects so as to obtain highly efficient, luminescent uMWCNTs (E-uMWCNTs). As an excellent “binder”, polyvinyl alcohol (PVA) has been wet-spun by lots of researchers [26,27]; here, we used a small amount of PVA as a matrix to prepare the macromorphic fibers of E-uMWCNTs. Due to the reinforcing effect of E-uMWCNTs on PVA-matrix, macromorphic E-uMWCNT fibers (E-uMWCNT-Fs), the high tensile strength of the fibers (228.2 MPa) was achieved. On the other hand, the E-uMWCNT-Fs have shown highly efficient photoluminescent properties, with a strong emitting intensities and long lifetimes. In particular, benefiting from the PVA-screening effect, E-uMWCNT-Fs exhibit excellent levels of media-resistance performance; they can maintain a strong UV-light response in water, high-concentration metal-ion solutions, and pH 4~12 aqueous solutions. These excellent photon quantum characteristics of the braidable, macromorphic fibers with high content levels of E-uMWCNTs show a wide range of potential applications, such as wearable fabrics, sensors, etc. in seriously polluted environments.

## 2. Experimental Section

### 2.1. Materials

Multiwalled carbon nanotubes (MWCNTs, inner diameter: 8~15 nm, outer diameter: 20~30 nm, length: 10~30 μm), Dimethyl sulfoxide (DMSO, AR), Europium(III) chloride hexahydrate (EuCl_3_·6H_2_O, 99.99%), 2-Thenoyltrifluoroacetone (TTA, 98%), 1,10-Phenanthroline monohydrate (98%), Chromium chloride hexahydrate (CrCl_3_·6H_2_O, AR), Manganese(II) chloride tetrahydrate (MnCl_2_·4H_2_O, AR), Acetic acid (C_2_H_4_O_2_, 99.5%) and Ammonium hydroxide solution (H_5_NO, 25%~28%) were purchased from Shanghai Macklin Biochemical Co., Ltd., Shanghai, China. Sulfuric acid (H_2_SO_4_, 98%), Sodium sulphate (Na_2_SO_4_, AR), Potassium chloride (KCl, AR), Magnesium chloride hexahydrate (MgCl_2_·6H_2_O, AR), Iron chloride hexahydrate (FeCl_3_·6H_2_O, AR), Cupric chloride dehydrate (CuCl_2_·6H_2_O, AR), and Cadmium sulfate (CdSO_4_, AR) were purchased from Sinopharm Chemical Reagent Co., Ltd., Shanghai, China. Poly(vinyl alcohol) (PVA, 98~99 mol%) and Cobalt nitrate hexahydrate (CoCl_3_·6H_2_O, AR) were purchased from Shanghai Aladdin Biochemical Technology Co., Ltd., Shanghai, China.

### 2.2. Electrochemical Method of Preparation of uMWCNTs

Unzipped, multiwalled carbon nanotubes (uMWCNTs) were prepared via an electrochemical method. Titanium sheets with a high level of purity (99.99%) were used as the electrode, and sulfuric acid diluted to 80% with deionized water was used as the electrolyte. A quantity of 1000 mg of original, multiwalled carbon nanotubes (MWCNTs) were weighed and spread evenly on a 10 cm × 10 cm titanium sheet, and they were then wrapped with PTFE film. Another titanium sheet of the same size was laid on top on the MWCNT-loaded titanium sheet. Then, the set-down “sandwich” of electrode/MWCNTs/electrode (ECE) was placed into the electrolytic cell, which was completely immersed in an 80% sulfuric acid solution. The first MWCNT-loaded titanium sheet served as the anode, and the other titanium sheet served as the cathode. The MWCNTs were electrochemically unzipped by applying 30 V of direct current for 8 h; then, the unzipped MWCNTs were washed with deionized water to pH = 7 and dried under vacuum at 80 °C for 12 h to obtain uMWCNTs (Figure 1a).

### 2.3. Preparation of Macromorphic Fibers by Wet-Spinning Method

*Preparation of macromorphic fibers of uMWCNT (uMWCNT-Fs)*: The macromorphic fibers formed by uMWCNT in PVA were designated as uMWCNT-Fs. The uMWCNTs (0 mg, 100 mg, 200 mg, 300 mg, 400 mg, and 500 mg) were added to 20 mL DMSO, and then the mixture was ultrasonically dispersed for 4 h. The abovementioned solutions were added to PVA particles (2000 mg, 1900 mg, 1800 mg, 1700 mg, 1600 mg, and 1500 mg) under heating and stirring at 95 °C. Next, the mixed solutions were stirred at 95 °C for 4 h so that they were completely dissolved, and then they were defoamed at 80 °C for 2 h. Finally, spinning solutions with different contents of uMWCNTs were obtained.

A supersaturated NaSO_4_ solution at 40 °C was used as a coagulation bath to wet-spin the above-prepared spinning dope with the model shown in Figure 1b, and we obtained 0 wt%, 5 wt%, 10 wt%, 15 wt%, 20 wt%, and 25 wt% of uMWCNT@PVA macromorphic fibers (uMWCNT-Fs), respectively. For comparison, 20 wt% MWCNT@PVA macromorphic fibers (MWCNT-Fs) were prepared by the same method.

*Preparation of E-uMWCNT macromorphic fibers (E-uMWCNT-Fs)*: According to the results in Section 3.2, the content of uMWCNTs in the macromorphic fibers prepared during this stage was 20 wt% (uMWCNT: PVA = 400 mg: 1600 mg), and the preparation method was the same as that for the uMWCNT-Fs. The steps were as follow: First, EuCl_3_·6H_2_O, 2-Thenoyltrifluoroacetone, and 1,10-Phenanthroline monohydrate were gradually added to DMSO at a molar ratio of 1:3:1 and stirred for 2 h to obtain 20 mg/mL Eu(TTA)_3_Phen-complex solution. The Eu(TTA)_3_Phen-complex solution and uMWCNTs were stirred for 6 h to form Eu(TTA)_3_Phen-complex-anchored uMWCNTs (E-uMWCNTs); then, the spinning solutions were prepared and spun according to different additions of Eu(TTA)_3_Phen (20 mg, 60 mg, 100 mg, 200 mg, 300 mg, and 400 mg). Finally, we obtained 1, 3, 5, 10, 15, and 20 wt% (contents of Eu^3+^-complexes in micromorphic fibers) E-uMWCNT@PVA micromorphic fibers (E-uMWCNT-Fs), respectively.

We then prepared 0.1 M Na_2_SO_4_, KCl, and MgCl_2_ solutions and 0.01 M of CoCl_2_, CrCl_3_, FeCl_3_, CuCl_2_, MnCl_2_, and CdSO_4_ solutions for the various metal-ion resistance testing of the E-uMWCNT-Fs. The acid–base resistance testing used aqueous solutions with pH = 4 and pH = 12, prepared with acetic acid and ammonia as the detection media.

### 2.4. Characterizations

The surface morphologies of MWCNTs, uMWCNTs, and E-uMWCNTs were carried out on a transmission electron microscope (TEM, JEOL, JEM-1200EX). The functional groups and structures of the MWCNTs and uMWCNTs were characterized using Fourier transform infrared spectroscopy (FT-IR, Thermo Scientific, Waltham, MA, USA, Nicolet iS10) and a Raman spectrometer (Thermo Scientific, Waltham, MA, USA, Nicolet 5700) equipped with a 532 nm laser source. The oxygen content of the uMWCNTs was measured using an element analyzer (EA, Elementar UNICUBE). Chemical reactions were measured via X-ray photoelectron spectroscopy (XPS, Thermo Scientific, Waltham, MA, USA, ESCALAB Xi+). The macromorphologies of the micromorphic fibers were observed via a stereo microscope (SZ780, Chongqing Optec Instrument, Chongqing, China); the micromorphologies and elemental-mapping of the micromorphic fibers were performed on a scanning electron microscope (SEM, TESCAN, VAGE 3). The thermal properties of the micromorphic fibers were measured using a thermo-gravimetric analyzer (TGA, HITACHI, STA7200) in a nitrogen atmosphere at a heating rate of 10 °C/min from 20 °C to 800 °C. The mechanical properties of the fibers were tested by a universal material-testing machine (Instron, 5300) at a speed of 200 mm/min and a 30 mm working-length after we stretched the primary fibers to four-times their original lengths. The luminescence properties of the macromorphic fibers were tested on a photoluminescence spectrometer (PL, Edinburgh, UK, FLS 1000).

## 3. Results and Discussion

### 3.1. Morphology and Structure of uMWCNTs

Figure 1a,c shows the TEM images of MWCNTs and uMWCNTs; it can be seen that the surface of the MWCNTs became rougher after the electrochemical unzipping reaction, and the average diameter of the uMWCNTs (Figure 1d, OD = 39.51 ± 0.59 nm) was larger than that of the MWCNTs (Figure 1b, OD = 26.30 ± 0.53 nm) by statistical calculations; meanwhile, the uMWCNTs showed better dispersibility in water after standing for 72 h (right side of Figure 1b,d).

To observe the changes on the surface of the MWCNTs after electrochemical unzipping reaction, we performed FT-IR, XPS, and Raman measurements on the MWCNTs and uMWCNTs. By comparing the FT-IR spectra of the uMWCNTs and MWCNTs (Figure 1e), it can be concluded that the uMWCNTs indicate an increased -C-O-C stretching vibration peak at ~1100 cm^−1^ and a -C=O stretching vibration peak at ~1700 cm^−1^, which can preliminarily indicate that more oxygen-containing groups were produced on the surface of the uMWCNTs after the electrochemical unzipping reaction. Based on the XPS spectra of the MWCNTs and uMWCNTs, the C1s spectra of the MWCNTs (Figure 1f) can be split into three sub-peaks: a C=C group, a C-OH group and a π-π* shakeup feature [28] at 284.9 eV, 286.5 eV, and 291.9 eV, while the C1s spectra of the uMWCNTs (Figure 1g) can be divided into four peaks, with the additional, new peak of a C=O group shown at 288 eV greater than that of the MWCNTs, which corresponds to the result of the FT-IR spectrum. Additionally, the oxygen-content of carbon nanotubes increased to 6.774% from 2.856% (Table 1) after the electrochemical unzipping reaction. Raman spectroscopy can be used to infer the surface structure of materials. By comparing the R(I_D_ / I_G_)(R_uMWCNT_ = 1.11 > R_MWCNT_ = 0.71) and 2D peak intensity (I_uMWCNT_ < I_MWCNT_) of the uMWCNTs and MWCNTs (Figure 1h), it can be concluded that the surfaces of the carbon nanotubes after electrochemical unzipping reaction were broken to a certain extent. Thus, based on the series of characterizations discussed above regarding MWCNTs and uMWCNTs, it can be fully demonstrated that the surfaces of the carbon nanotubes were partially unzipped and generated a large number of oxygen-containing groups after the electrochemical unzipping reaction. Thus, the dispersion and surface activity of the carbon nanotubes were improved for use in further applications.

### 3.2. Structures and Performances of uMWCNT-Fs

Microscopic pictures and SEM images shows the morphological structures of the pure PVA fibers and macromorphic fibers of the uMWCNTs (uMWCNT-Fs), respectively. It can be seen from the microscopic pictures of the pure PVA fibers (Figure 2a) and uMWCNT-Fs (Figure 2e) that the uMWCNT-Fs lose their transparence due to the addition of the uMWCNTs, and the fibers’ surfaces become rough in comparison with an SEM image of a pure PVA fiber (Figure 2b) and uMWCNT-Fs (Figure 2f). Additionally, the cross-section of the uMWCNT-F (Figure 2g,h) shows dense, protruding images that are surely distributed by the uMWCNTs, while the cross-section of the pure PVA fiber (Figure 2c,d) is quite smooth.

The tensile properties of uMWCNT-Fs with different contents of uMWCNTs were measured and are shown in Figure 3a,b and Table 2. The results show that the macromorphic fibers composed of 20 wt% uMWCNTs (tensile strength = 183.7 MPa) have the highest tensile strength, which is 29.3% higher than that of pure PVA fiber (tensile strength = 142.1 MPa). Thus, we used 20 wt% as the optimal content of uMWCNTs in macromorphic fibers in further work. In addition, the actual content of the uMWCNTs in the macromorphic fibers can be found in the TGA data (Figure 3c). In Figure 3c, both pure PVA fibers and uMWCNT-F macromorphic fibers began to lose weight from 100 °C due to the volatilization of crystalized water. Then, in the temperature range of 120~260 °C, slight weight losses were caused by the side condensation of the PVA. At about 318 °C, dramatic weight losses happened, caused by PVA degradation, and finally the weight loss remained at around 480 °C. The pure PVA fibers and the 20 wt% uMWCNT-Fs had residues of 2.9 wt% and 22.7 wt% after calcination, which indicated the actual content of the uMWCNTs in the macromorphic fibers of the 20 wt% uMWCNT-Fs. In addition, it can be seen from the DTG curves (Figure 3d) that the pyrolysis rate of the pure PVA fiber is higher than that of uMWCNT-F in almost every temperature range, which reflects that uMWCNTs improved the heat resistance of the fiber.

### 3.3. Structures and Performances of E-uMWCNT-Fs

The morphological structure and the interaction between the Eu^3+^-complex and the uMWCNTs of the E-uMWCNTs were characterized via TEM and XPS, respectively. As we can see from the TEM image of the E-uMWCNTs, the Eu(TTA)_3_phen-complexes were uniformly anchored on the surface of the uMWCNTs (Figure 4a).

By dividing the O1s spectra of the Eu(TTA)_3_phen-complexes (Figure 4b) and E-uMWCNTs (Figure 4c), it can be found that they both indicated the Eu-O bond peak around 532 eV, which is a covalent bond created by the combination of Eu^3+^-ions with oxygen ions in organic ligands. While the E-uMWCNTs had a lager Eu-O peak area, which means that the Eu^3+^-ions in E-uMWCNTs had stronger interactions with the oxygen-containing groups on the surfaces of the uMWCNTs. Here, we regard this special structure as a “mortise and tenon” structure, while there is an anchoring effect between them. Previously, Qi et al. [29] also reported a Eu^3+^-complex/carbon nanotube hybrid material with a similar structure, but the structure of the carbon nanotubes mentioned above was severely damaged, having failed to present the macromorphic properties. Here, we obtained the wet-spinning fibers from E-uMWCNTs, which not only showed excellent luminescence properties, but they also exhibited the excellent mechanical properties of macromorphic morphology.

The morphological structures of the macromorphic fibers of the E-uMWCNTs (E-uMWCNT-Fs) are shown in Figure 5a. It can be seen that fiber surface is rough (Figure 5b), and there are a large number of uMWCNTs that are uniformly erect at the cross-section (Figure 5c,d) of E-uMWCNT-F. The SEM elemental-mapping images of E-uMWCNT-Fs (Figure 5e–h) show that the Eu(TTA)_3_phen-complexes are distributed uniformly in the cross-sections of the macromorphic fibers.

The mechanical properties of the E-uMWCNT-Fs with different contents of Eu(TTA)_3_phen-complexes are shown in Figure 6a,b and Table 3. It can be seen that, due to the addition of E-uMWCNTs, the mechanical properties of the macromorphic fibers have been greatly improved, in which the tensile strength of the 15 wt% E-uMWCNT-F (tensile strength = 228.2 MPa) is 60% higher than that of the pure PVA fiber, and it is also 24% higher than that of the uMWCNT-F macromorphic fibers. According to our previous research], it can be concluded that the surfaces of the E-uMWCNTs have a rivet-like structure, resulting in an interface-enhancement effect between the E-uMWCNTs and the PVA host.

On the other hand, the photon quantum characteristics of the E-uMWCNT-Fs are shown in Figure 7. The spectrum of E-uMWCNT-F is indicated in Figure 7a, from which we chose an excitation wavelength of 374 nm, and thus, the emission spectrum was collected, as shown in Figure 7b, from which we found that the maximum emission peak was 615 nm, with 3 obvious emission bands at 590 nm, 650 nm, and 700 nm. There is an antenna effect between the Eu^3+^ ions and the organic ligands (2-Thenoyltrifluoroacetone (TTA) and 1,10-Phenanthroline monohydrate (Phen)) when they form a complex. The organic ligands (both TTA and Phen) act as antennae to fully accept the light energy under the excitation at 374 nm, and then the excitation energy of the pi-electrons will transfer to the f-electrons in the f-electron orbital of the Eu^3+^ ions. Eventually, the complex emits visible red light corresponding to the f-f electronic transition of Eu^3+^ ions. The Eu^3+^ ion emission features of E-uMWCNT-Fs arise from ^5^D_0_→^7^F_J_ (J = 0-4) f-f electronic transitions corresponding to ^5^D_0_→^7^F_2_ (615 nm), ^5^D_0_→^7^F_1_ (590 nm), ^5^D_0_→^7^F_3_ (650 nm), and ^5^D_0_→^7^F_4_ (700 nm) transitions [30,31]. In this case, under the excitation of light energy at 374 nm, the organic ligands in the Eu^3+^-complex first undergo π-π* absorption, that is, through the electronic transition from single ground state to single excited state (S→S*), and then through the intersystem crossing (ISC) to triple excited state (T1); the energy is transferred from the T1 to the energy level of the f-electrons of the Eu^3+^ ions, and the luminescence is generated by the radiation transition from the high vibrational energy level to the low vibrational energy level [32]. The luminescence lifetime shown in Figure 7c is indicated as 479 µs, and this longer lifetime of excited 4f electrons causes the E-uMWCNT-Fs to have many sensitive indications for sensing the influence factors [33]. In addition, we measured that the quantum yield of the E-uMWCNT-Fs, which were typically between 11–15% (Table 4).

As they are excited at 374 nm, the variety of photon quantum characteristics of E-uMWCNT-Fs corresponding to different contents of Eu(TTA)_3_phen anchored at 20 wt% uMWCNTs in E-uMWCNT-Fs (Figure 8a), we found that the luminescence intensities increased with the additional content increase of Eu(TTA)_3_phen-complexes. To investigate the sensitivity of the photon quantum characteristics of the E-uMWCNT-Fs, we organized a group of experiments to sense the effects of pH, water, non-heavy-metal ions, and heavy-metal ions. As shown in Figure 8b, we found that the emissions at 615 nm (excited at 374 nm) retained almost no changes of the E-uMWCNT-Fs’ emission intensity, which confirms that water has no obvious quenching effect on the emission of E-uMWCNT-Fs, leading to the conclusion that E-uMWCNT-Fs are water-resistant. For the metal-ion influence on the emission of E-uMWCNT-Fs, we finished two groups of metal ions: non-heavy-metal ions and heavy-metal ions. The results (Figure 8e,f) show that non-heavy-metal ions had no obvious quenching effect either, whereas heavy-metal ions had obvious quenching effects. However, the E-uMWCNT-Fs still showed high heavy-metal ions resistance when compared to the Eu(TTA)_3_phen-complexes soaking in heavy-metal-ions solutions for shorter times (shown in Figure 8c,d). The luminescence intensity decreases verified the dynamic quenching mechanism of the transition-metal ions on E-uMWCNT-Fs because the coming Cu^2+^ or Fe^3+^ ions were able affect the interactions of the Eu^3+^ ions with oxygen atoms in the TTA and nitrogen atoms in the Phen, further affecting energy-transfer efficiency between the ligands and the Eu^3+^ ions; thus, the interference led to luminescence quenching [34,35,36]. Additionally, E-uMWCNT-Fs can become efficiently photoluminescent in heavy-ion solutions due to the larger number of Eu^3+^-complexes that were loaded on the surfaces of the uMWCNTs, and the E-uMWCNTs were coated with PVA, which prevented contact between the E-uMWCNTs and the heavy-metal ions. Meanwhile, the results in Figure 8f also show that the E-uMWCNT-Fs retained a good level of photoluminescence properties at pH 4 and 12 when the soaking time was maintained similarly to that used in the different heavy-metal ions.

The high-efficiency photoluminescence properties of E-uMWCNT-Fs are vividly demonstrated in its fabric (Figure 9a), which emit a bright, red luminescence under UV light when immersed in various liquid media, as shown in Figure 9b. E-uMWCNT-F fabric shows a red luminescence of a different intensity under UV light in various liquid media, which is consistent with the data, as shown in Figure 8e,f.

## 4. Conclusions

In summary, we used an improved electrochemical methodology to functionalize the original MWCNTs, resulting in partially unzipped MWCNTs (uMWCNTs) with a large number of oxygen-containing groups on their surfaces. Subsequently, we prepared a macromorphic fiber with a high content of uMWCNTs by wet spinning, and we determined that the highest tensile modulus (183.7 MPa, 30% higher than that of pure PVA fibers) was obtained when the content of MWCNTs in the PVA matrix reached 20 wt%. The surface defects and functional groups of the uMWCNTs were sufficient for use in preparing an Eu^3+^-complex-anchored uMWCNTs–E-uMWCNTs luminescent hybrid material with a special surface structure. Eventually, a macromorphic fiber with numerous excellent properties was prepared by using this novel, luminescent hybrid material. The macromorphic fiber containing 30 wt% of E-uMWCNTs has a tensile modulus (228.2 MPa) that is 1.6 times greater than that of pure PVA fiber. Meanwhile, E-uMWCNT-F has excellent luminescence properties with a long luminescence lifetime (479 μs) and a better quantum yield (11–14%) due to the f-f electron transition of Eu^3+^ ions. It can emit a dazzling red light under the excitation of UV light. Moreover, these braidable fibers exhibit high-efficiency photoluminescence in water and in non-heavy-metal-ion solutions. Additionally, the fibers can maintain a favorable UV-light response of high-quality photoluminescence—even in heavy metal solutions, acidic solutions (pH > 4), and alkaline solutions (pH < 12), depending on the special f-f electron transitions of the E-uMWCNTs and the PVA-coating effect on the E-uMWCNTs.

## Data Availability

Not applicable.

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
