# Peer review of "Excellent Photonic and Mechanical Properties of Macromorphic Fibers Formed by Eu3+-Complex-Anchored, Unzipped, Multiwalled Carbon Nanotubes"

_materials, 2022, doi:10.3390/ma15144933_

Round 1
Reviewer 1 Report
The synthesis of Eu3+-complex-anchored unzipped multi-walled carbon nanotubes is reported in this work. The average diameter of nanofibers is about 20-40 nm. Th obtained materials possess the highest tensile modulus (183.7 MPa, 30% higher than that of pure PVA fibers). Eu(TTA)3phen-modified fibers demonstrate noticeable luminescence corresponding to the f-f transitions of Eu3+ ion upon 374-nm excitation. Addition the heavy metal ions results in luminescence quenching. This work in principal can contribute to the Materials sciences and Analytical chemistry (sensors design). In principle, manuscript can be accepted after addressing the following comments:
1) Fig. 6. captures: please add the description of the curves a-g in Fig 6a.
2) Lines 272-281: The excitation spectra have the maximum at the 374 nm. Please assign this maximum to the certain electronic transition. Probably, this maximum corresponds to the pi-pi* transition of the ligand or carbon fiber. In this case the reported material demonstrates the antenna effect. Please, discuss the photophysical mechanism in details.
3) Lines 290-307: Please explain the mechanism of the luminescence quenching by the heavy metal ions.
4) Lines 311-313: Fig 9b shows the difference in emission spectra/color of fabric made from the reported E-uMWCNT-Fs fibers soaked in water and aqueous solution of Na+/K+. This observation should be discussed.
5) Could you estimate the quantum yield of luminescence of the reported materials?
Author Response
First of all, we would like to thank the Reviewer for the high evaluation for our work: “This work in principal can contribute to the Materials sciences and Analytical chemistry (sensors design)”.
Point 1: Fig. 6. captures: please add the description of the curves a-g in Fig 6a.
Response 1: The sample names of a-g in Figure 6 a. have been tabulated and posted in the manuscript, see Table 3 in the manuscript for details.
Point 2: Lines 272-281: The excitation spectra have the maximum at the 374 nm. Please assign this maximum to the certain electronic transition. Probably, this maximum corresponds to the pi-pi* transition of the ligand or carbon fiber. In this case the reported material demonstrates the antenna effect. Please, discuss the photophysical mechanism in details.
Response 2: Thank you very much fo your nice question.
Firstly, the maximum at the 374 nm of excitation wavelength is surely to be caused by pi-pi* transition from ligand that has the pi-pi conjugate structure. You are right.
Secondly, we added the analyses of emission spectra, in text as below: The Eu(III) emission features of E-uMWCNT-Fs arise from 5D0→7FJ (J = 0-4) f-f electronic transitions, and the energy gaps between ground states and excited states, corresponding to 5D0→7F2 (615 nm), 5D0→7F1 (590 nm), 5D0→7F3 (650 nm) and 5D0→7F4 (700 nm) transitions.
Point 3: Lines 290-307: Please explain the mechanism of the luminescence quenching by the heavy metal ions.
Response 3: The heavy metal ions are combined with the framework of the complexes’ ligand when Eu3+-complexes are mixed with the heavy metal ions in solutions, thus, the excitation energy is transferred to the heavy metal ions which causes the complexes fluorescence quenching. And it is also supplemented in the manuscript.
Point 4: Lines 311-313: Fig 9b shows the difference in emission spectra/color of fabric made from the reported E-uMWCNT-Fs fibers soaked in water and aqueous solution of Na+/K+. This observation should be discussed.
Response 4: The E-uMWCNT-F’s fabrics shows red fluorescence in different intensity under UV light in various liquid media because of the different photoluminescence resistance of E-uMWCNT-F to different ions. And it is also been discussed in the manuscript.
Point 5: Could you estimate the quantum yield of luminescence of the reported materials?
Response 5: The quantum yield of the E-uMWCNT-Fs measured by Photoluminescence Spectrometer were between 11-15%. And it is also supplemented in the manuscript.
Reviewer 2 Report
Report on Excellent Photonic and Mechanical Properties of Macromorphic Fibers formed by Eu3+-Complex-Anchored Unzipped Multiwalled Carbon Nanotubes by Mengjie Huang et al. The manuscript details the use of the electrochemical method to tear the surface of multi-walled carbon nanotubes (MWCNT) to anchor photonic Eu3+-complex and by wet spinning process obtaining long fibres. The manuscript is well structured and describes step-by-step the characterisation process. The discussion of results is brief and needs to include the physical phenomena behind them. The conclusions are adequate and summarise the experimental findings. After addressing the following issues, the manuscript would be acceptable for publication in the Materials journal.
1. There is a great similitude in experimental techniques and results on the same materials between this manuscript and the other previously published by the corresponding author: https://doi.org/10.1016/j.jallcom.2020.156880 Please discuss it. I advise you to include it in the references after that.
2. OD values in lines 175-176 are misplaced.
3. Figure 3 a. Please disclose contents corresponding to the labels a,..,g
4. Lines 299-305, I think this is the main contribution to the photon quantum characteristics of E-uMWCNT-F. However, there is no physical explanation or a phenomenological model to explain the results. Please include it and comment on the conclusions.
Author Response
First of all, we would like to thank the Reviewer for the high evaluation for our work: “The manuscript is well structured and describes step-by-step the characterisation process. The conclusions are adequate and summarise the experimental findings”.
Point 1: There is a great similitude in experimental techniques and results on the same materials between this manuscript and the other previously published by the corresponding author: https://doi.org/10.1016/j.jallcom.2020.156880 Please discuss it. I advise you to include it in the references after that.
Response 1: Previously, Qi et al. (https://doi.org/10.1016/j.jallcom.2020.156880) reported a Eu3+-complex/carbon nanotube hybrid material with a similar structure, but the structure of the carbon nanotubes they mentioned was severely damaged, and they failed to present the macromorphic properties of this hybrid material. Here, we obtained the wet-spinning fibers of E-uMWCNTs, which not only show excellent fluorescence properties, but also exhibit excellent mechanical properties in macromorphic morphology. And it is also been discussed in the manuscript.
Point 2: OD values in lines 175-176 are misplaced.
Response 2: Mislabeled OD values in the manuscript have been corrected.
Point 3: Figure 3 a. Please disclose contents corresponding to the labels a,..,g
Response 3: The sample names of a-g in Figure 3 a. have been tabulated and posted in the manuscript, see Table 2 in the manuscript for details.
Point 4: Lines 299-305, I think this is the main contribution to the photon quantum characteristics of E-uMWCNT-F. However, there is no physical explanation or a phenomenological model to explain the results. Please include it and comment on the conclusions.
Response 4: The heavy metal ions are combined with the framework of the complexes’ ligand when Eu3+-complexes are mixed with the heavy metal ion solutions, thus, the excitation energy is transferred to the heavy metal ions which causes the complexes fluorescence quenching. And E-uMWCNT-Fs can be efficiently photoluminescence in heavy ion solutions due to a larger number of Eu3+-complexes were loaded on the surface of uMWCNT and E-uMWCNTs were coated with PVA that avoids the contact between E-uMWCNTs and heavy metal ions. And it is also been discussed in the manuscript.
Round 2
Reviewer 1 Report
Authors improved the manuscript and partially addressed my comments. However, some questions were not answered correctly. Please, correct the manuscript by addressing the followings points:
1) Point 2: Lines 287-299: The authors assigned 374-nm excitation to pi-pi* transition of the conjugated ligand. The emission bands correspond to the f-f transitions of Eu3+ ions. I strongly suggest to provide the photophysical mechanism indicating the energy transfer processes. Otherwise, it reads strange that upon excitation of the ligand, one observes luminescence from Eu3+ ion.
2) Point 3: Lines 323-329: Authors actually did not explained the mechanism of the luminescence quenching by the heavy metal ions. Authors stated that “the excitation energy is transferred to the heavy metal ions which causes the complexes fluorescence quenching”. Which electronic state is quenched by the heavy metal ions?
3) All over the manuscript authors called the observed photoluminescence as “fluorescence”. However, the observed luminescence corresponds to the 5D0→7FJ transitions of Eu3+ ion, where the multiplicity changes, and therefore can not be called “fluorescence”. Please, correct.
Author Response
Thank you for your revision, the following is my reply.
Point 1: Lines 287-299: The authors assigned 374-nm excitation to pi-pi* transition of the conjugated ligand. The emission bands correspond to the f-f transitions of Eu3+ ions. I strongly suggest to provide the photophysical mechanism indicating the energy transfer processes. Otherwise, it reads strange that upon excitation of the ligand, one observes luminescence from Eu3+ ion.
Response 1: Thank you very much for your good suggestion. We have made the discussion as below and inserted into manuscript from line 293-307:
There is an antenna effect between Eu3+ ions and organic ligands (2-Thenoyltrifluoroacetone (TTA) and 1,10-Phenanthroline monohydrate (Phen)) when they form a complex. The organic ligands (both TTA and Phen) act as antenna to fully accept the light energy under the excitation at 374 nm, and then the excitation energy of pi-electrons will transfer to f-electrons in f-electron orbital of Eu3+ ions. Eventually, the complex emits visible red light corresponding to the f-f electronic transition of Eu3+ ions. The Eu3+ ions emission features of E-uMWCNT-Fs arise from 5D0→7FJ (J = 0-4) f-f electronic transitions corresponding to 5D0→7F2 (615 nm), 5D0→7F1 (590 nm), 5D0→7F3 (650 nm) and 5D0→7F4 (700 nm) transitions. In this case, under the excitation of light energy at 374 nm, the organic ligands in the Eu3+-complex first undergo π-π* absorption, that is, through the electronic transition from single ground state to single excited state (S→S*), and then through the intersystem crossing (ISC) to triple excited state (T1), the energy is transferred from the T1 to the energy level of f-electrons of Eu3+ ions, and the luminescence is generated by the radiation transition from the high vibrational energy level to the low vibrational energy level.
And added the quantum yield data in Table 4, with the description from line 310-311, as below:
In addition, we also measured that the quantum yield of E-uMWCNT-Fs are typically between 11-15% (Table 4.).
Point 2: Lines 323-329: Authors actually did not explained the mechanism of the luminescence quenching by the heavy metal ions. Authors stated that “the excitation energy is transferred to the heavy metal ions which causes the complexes fluorescence quenching”. Which electronic state is quenched by the heavy metal ions?
Response 2: Thank you very much for your comments. We added the possible mechanism discussion into the manuscript, between line 335-340, as below:
The luminescence intensity decreases verified the dynamic quenching mechanism of the transition metal ions on E-uMWCNT-Fs , because the coming Cu2+ or Fe3+ ions could affect the interactions of Eu3+ ions with oxygen atoms in TTA and nitrogen atoms in Phen, and further affect the energy transfer efficiency between the ligands and Eu3+ ions, thus the interference leads luminescence quenching.
Point 3: All over the manuscript authors called the observed photoluminescence as “fluorescence”. However, the observed luminescence corresponds to the 5D0→7FJ transitions of Eu3+ ion, where the multiplicity changes, and therefore can not be called “fluorescence”. Please, correct.
Response 3: Thank you, it is a good question. Yes, you are right. We have corrected through out this manuscript, luminescence replaced fluorescence.

Reviewer 2 Report
The authors did an excellent job fulfilling my concerns in all comments, except they did not include the conclusions section on the phenomenological model insights of photon quantum characteristics of E-uMWCNT-F. Nevertheless, I believe that information is relevant enough to be mentioned.
A piece of advice, please use the line numbers to address each of the corrections, amendments, suggestions, etc. It is crucial to point out what and how you are answering them.
Author Response
Thank you for your revision, the following is my reply.
Point 1: The authors did an excellent job fulfilling my concerns in all comments, except they did not include the conclusions section on the phenomenological model insights of photon quantum characteristics of E-uMWCNT-F. Nevertheless, I believe that information is relevant enough to be mentioned.
Response 1: Thanks for your advice. Following your suggestion, we have listed the quantum yields of E-uMWCNT-Fs (page 9, Table 4.) (between line 287-288). And we have discussed the photon quantum characteristics in the conclusion section, as below (line 366-373 on page 11-12) :
Meanwhile, E-uMWCNT-F has excellent luminescence properties with a long luminescence lifetime (479 μs) and better quantum yield (11-14%) due to the f-f electron transition of Eu3+ ions, it can emit dazzling red light under the excitation of UV light. Moreover, this braidable fibers exhibit high-efficiency photoluminescence in water and in non-heavy metal ion solutions. Also, the fibers can maintain a favorable UV light response to behave high quality photoluminescence even in heavy metal solutions, acidic solutions (pH > 4) and alkaline solutions (pH < 12), depending on the special f-f electron transitions of E-uMWCNTs and the PVA coating effect on E-uMWCNTs.

Round 3
Reviewer 1 Report
Authors improved the manuscript and addressed my comments. The manuscript can be accepted in present form.